# DENSELY CONNECTED RECURRENT NEURAL NETWORK FOR SEQUENCE-TO-SEQUENCE LEARNING

## ABSTRACT

Deep neural networks based sequence-to-sequence learning has achieved remarkable progress in applications like machine translation and text summarization. However, sequence-to-sequence models suffer from severe inefficiency in training process, requiring huge amount of training time as well as memory usage. In this work, inspired by densely connected layers in modern convolutional neural network, we introduce densely connected sequence-to-sequence learning mechanism to tackle this challenge. In this mechanism, multiple layers of representations from stacked recurrent neural networks are concatenated to enhance feature reuse. Furthermore, a densely connected attention model is elaborately leveraged to improve information flow with more efficient parameter usage via multi-branch structure and local sparsity. We show that such a densely connected mechanism significantly reduces training time and memory usage for sequence-to-sequence learning. In particular, in WMT-14 English-French translation task with a subset of $12M$ training data, it takes half of training time and model parameters to achieve similar BLEU as typical stacked LSTM models.

## 1 INTRODUCTION

Deep neural networks based sequence-to-sequence (S2S) learning has made rapid progress in recent years, for several real-world applications, such as neural machine translation (NMT) (Sutskever et al., 2014; Bahdanau et al., 2014), text summarization (Nallapati et al., 2016) and semantic parsing (Vinyals et al., 2015). S2S learning takes an encoder-decoder framework (Cho et al., 2014b; Sutskever et al., 2014): The encoder takes source sequence as input and generates a set of hidden representations, based on which the decoder generates the target-side sequence. To enhance the information flow between the encoder and the decoder, attention mechanisms (Bahdanau et al., 2014; Luong et al., 2015) have been introduced to model the dynamic dependency between the tokens in the source and target side sequences.

To better model the complicated relationships between source and target side sequences, the S2S approach is leveraging deeper and deeper neural network architectures. For example, the stacked long-short term memory (LSTM) based model used in Google's Neural Machine Translation system (GNMT) (Wu et al., 2016) has 8 layers for both the encoder and the decoder; similar model complexity is observed in convolutional S2S (ConvS2S) (Gehring et al., 2016) learning, containing 30 layers in both sides for English-French translation. Such a large model complexity limits the scalability of modern S2S architectures. One of the most severe issues is training cost: 96 GPUs/6 days and 8 GPUs/38 days are needed for GNMT and ConvS2S respectively to train a model for English-French translation.

In modern computer vision literature where deep learning first flourishes, several model innovations make the training of very deep neural networks very effective (Szegedy et al., 2015; Ioffe & Szegedy, 2015; He et al., 2016). In this paper, we investigate the possibility of leveraging one of the most recent introduced methods, i.e., the dense connections used in DenseNet (Huang et al., 2016) to alleviate the sufferings of the S2S scalability issue. Through the dense connections between the current layer and all its preceding layers, the information flow through the deep convolutional networks gets enhanced and the extracted features in every layer are further reused in higher layers, making it possible to significantly reduce model parameters.

We incorporate dense connections into recurrent neural network (RNN) based S2S learning, which suffers most from poor scalability due to the recurrent nature in computation. Our model innovation mainly includes two folds: First, for the 2D computational process in the stacked RNN structure, i.e., the *horizontal* computation along time axis and the *vertical* computation along stacked deep axis, we add dense connections in the latter one. As a result, the input of each layer is the concatenation of representations from all its preceding layers. Such low-level feature re-usage not only reduces the *vertical* computation cost, but also the *horizontal* cost, given that the dimension of the hidden states to be calculated at next time step has reduced. Second, we additionally introduce the dense connections into attention computation, by connecting every encoder layer and the decoder layer in which attention is computed, and then concatenating the hidden states respectively computed to represent the source sequence in a more compact way. Acting in this way further reduces the parameters used in traditional attention mechanism.

We test the performance of our proposed densely connected recurrent networks on the WMT-14 English-German translation task, a subset of English-French translation task and a subset of Gigaword text summarization. Our results demonstrate that our densely connected RNNs outperform stacked RNNs when both use similar number of parameters. Moreover, when using much less parameters, e.g., half of parameters of original stacked version, densely connected RNN achieves comparable or even better performance with much less training time and memory cost.

## 2 BACKGROUND

### 2.1 RNN BASED SEQUENCE-TO-SEQUENCE LEARNING

In this section, we briefly describe the sequence-to-sequence (S2S) learning framework based on stacked recurrent neural networks and attention mechanism (Bahdanau et al., 2014). An RNN based S2S model typically contains an RNN encoder and an RNN decoder with attention mechanism as a bridge of the two parts. The overall architecture is shown in Fig. 1.

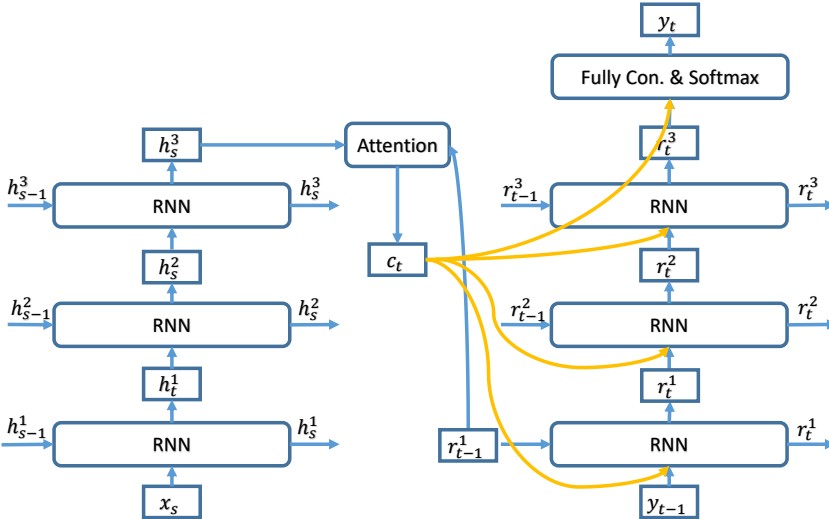

Figure 1: A typical RNN based encoder-decoder framework for S2S. The first layer in the decoder is used to compute the attention and the attentional context vectors are fed into every upper decoder layer.

**Encoder** The RNN encoder reads the input sequence $X = \{x_1, \cdots, x_N\}$ and maps it to a sequence of hidden states $\{h_1, h_2, \ldots, h_N\}$. To enhance model capacity, the encoder typically employs a stacked structure of RNN containing $L_e$ layers, in which the hidden states of lower-level layer are fed into the adjacent higher-level layer as input. Mathematically speaking, for the RNN layer $l$, its

hidden state $h_s^l$ at time step s is computed as:

$$h_s^l = g(h_{s-1}^l, h_s^{l-1}), \tag{1}$$

where $h_{s-1}^l$ is the hidden state of layer $l$ at the previous time step, and $h_s^{l-1}$ is the hidden state of the previous layer at the same time step ($h_s^1$ is the input embedding $x_s$). $g(\cdot)$ is the nonlinear function in the recurrent unit, which can be implemented in the vanilla form:

$$g(h, x) = f(W_{hh}h + W_{xh}x + b), \tag{2}$$

where $f$ is a non-linear activation function, or Long Short-Term Memory (LSTM) (Hochreiter & Schmidhuber, 1997), or Gated Recurrent Unit (GRU) (Cho et al., 2014a; Bahdanau et al., 2014).

**Decoder** Based on the source hidden states outputted by the top layer of the encoder, the decoder predicts a target sequence $Y = \{y_1, y_2, \ldots, y_M\}$ according the probability chain rule: $P(Y|X) = \prod_{t=1}^M P(y_t|y_{<t}, X)$. The generation probability of each single word $y_t$ is specified as:

$$P(y_t|y_{<t}, X) = softmax(y_{t-1}, c_t, r_t^{L_d}), \tag{3}$$

where $r_t^{L_d}$ is the hidden state outputted by the top layer (i.e., the $L_d$-th layer) of the decoder. That is to say, the encoder is also in a stacked form of multiple RNN layers and the hidden state $r_t^i$ of decoder layer $i$ at time step $t$ is computed as: $r_t^i = g(r_{t-1}^i, r_t^{i-1}, c_t)$, similar with the encoder but with the only difference in that an additional context vector $c_t$ is further provided as input. $c_t$ is used to dynamically represent the source-side information in decoding at the $t$-th time step and is computed through the attention mechanism:

$$\alpha_{st} = \frac{\exp(e_{st})}{\sum_{n=1}^N \exp(e_{nt})}, c_t = \sum_{s=1}^N \alpha_{st} h_s^{L_e}. \tag{4}$$

Here $\alpha_{st}$ models how important the source side information $h_s^{L_e}$ is to the $t$-th step decoding. $e_{st} = align(h_s^{L_e}, r_{t-1}^{l'})$ is typically implemented as a feed-forward neural network, where $l'$-th layer in the decoder is used to compute the attention (typically $l' = 1$).

## 2.2 DENSE CONVOLUTIONAL NEURAL NETWORK

Huang et al. (2016) introduces Dense Convolutional Neural Network (DenseNet). In DenseNet, every two layers are connected with each other to enhance the effectiveness of information flow. Different with ResNet (He et al., 2016) where residual is used as the connection operator, the connection in DenseNet is implemented as the concatenation of each layer, where the current layer obtains additional inputs (i.e., feature maps) from all its preceding layers, and concatenates them together as the input for subsequent computation. In this way, the low-level features in the very deep neural network are also used in the high-level computation, achieving highly efficient feature re-use and less model parameters.

Mathematically speaking, we denote the output (a.k.a. feature maps) of the $l$-th layer as $x_l$ and suppose there are $L$ layers in total. In DenseNet, the $l$-th layer receives the output of all its preceding convolutional layers $\{x_0, \ldots, x_{l-1}\}$, as its input:

$$x_l = H_l([x_0; x_1; \ldots; x_{l-1}]), \tag{5}$$

where $[x_0; x_1; \ldots; x_{l-1}]$ refers to the concatenation of $\{x_0, x_1, \cdots, x_{l-1}\}$ and $H_l(\cdot)$ is a composite non-linear operator such as batch normalization (Ioffe & Szegedy, 2015), rectified linear units (ReLU) (Glorot et al., 2011), pooling (Hariharan et al., 2015), or convolution. $H_l$ is further used to compute the feature maps of the $l$-th layer $x_l$.

## 3 MODELS

In this section, we describe the architecture of DenseRNN and discuss how dense connections can help to achieve highly efficient parameter usage for sequence-to-sequence learning. Dense connections are mainly leveraged in two parts: 1) the stacked RNN models in the encoder and decoder, i.e., intra connections between different layers within the same deep RNN, introduced in Subsection 3.1; 2) the attention models, i.e., inter connections between encoder layers and decoder layers in attention computation, introduced in Subsection 3.2. Fig. 2 shows the overall architecture of our proposed model.

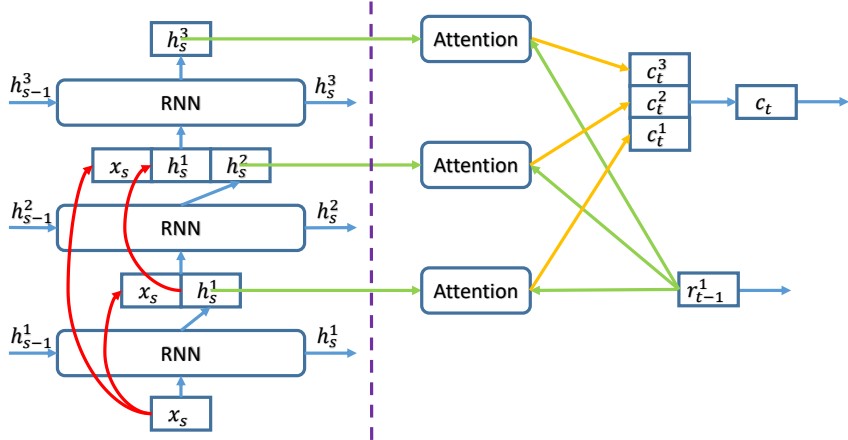

Figure 2: Illustration of our proposed DenseRNN for S2S learning. The figure is split by the purple dash line into two parts: 1) The left part indicates the dense encoder, where the red lines indicate dense connections, and the dense decoder has the similar architecture. 2) The right part indicates the dense attention mechanism, specified by the green and the yellow lines.

### 3.1 DENSE ENCODER AND DECODER

The stacked structure of deep RNN contains a two-dimensional (2D) computational process for the output $h_t^l$ of a particular unit: The recurrent computation along horizontal axis, and the feedforward computation along the vertical axis. Out of the two computational directions, we choose to add dense connections to the vertical axis, considering the following reasons: (1) Along the time axis, the ability of dynamically selecting input information provided at every time step and forgetting historical states is important, as indicated by the designing principle of LSTM and empirical verifications for the importance of forget gate (Greff et al., 2017). Therefore, adding dense connections along time axis might be a brute-force way to compel information flow, hurting the adaptiveness of modern RNN architectures such as LSTM and GRU. (2) As a comparison, the horizontal computation along $y$ axis connects the single-input/single-output, which might suffer from in-sufficient information flow. We still use $h_t^l$ to represent the output of the recurrent unit at the $t$-th timestep of layer $l$. Different with Eqn. 1, the recurrent unit will now accept the concatenated outputs from both $x$ and $y$ axis:

$$h_t^l = g(o_{t-1}^l, o_t^{l-1}) = g([h_{t-1}^0; \cdots; h_{t-1}^l], [h_t^0; \cdots; h_t^{l-1}]). \tag{6}$$

$[;]$ still denotes the concatenation operators for several vectors and $h_t^0 = x_t$. That is, apart from $h_t^l$, each unit is now associated with a new state vector $o_t^l$, computed by the concatenation of $h$ from all its bottom units and itself ($\leq l, t$). $o_t^l$ is further provided to the subsequent computation in both axes. Acting in this way, each layer in the stacked RNN architectures are connected with all its bottom layers through concatenation, making full use of low-level features that have been generated before. Significant parameter reduction is achieved via the dense connection implemented in this way, not only in the $y$ axis where dense connection is added but also in the $x$ axis containing each time-step. To see this, suppose the $g$ function is in the similar vanilla form as in Eqn. 2: $h_t^l = f(W_{hh}^l o_{t-1}^l + W_{xh}^l o_t^{l-1} + b)$. Assume that $o_{t-1}^l$ and $o_t^{l-1}$ keep the same dimension with the input of the vanilla RNN, then when the output $h_t^l$ is reduced $D$ times due to dense connections, the parameters in both $W_{hh}^l$ and $W_{xh}^l$ are reduced by a fraction of $1/D$ compared with that in Eqn. 2.

Such dense connections are added to the stacked RNN structure in both the encoder and the decoder side for S2S learning. Furthermore, in the decoder side, we also incorporate dense connections in the attention computation, which will be introduced in the next subsection.

## 3.2 DENSE ATTENTION

As shown in subsection 2.1, the attention score is typically computed based on the output of the last layer in the encoder $\{h_1^{L_e}, \cdots, h_N^{L_e}\}$. When the dense connection is added in the encoder side as introduced before, we actually have additional output vectors $\{o^{L_e}, \cdots, o_N^{L_e}\}$ for the top layer in the encoder. Each $o_s^{L_e}$ is the concatenation of the all the encoder layers' output: $o_s^{L_e} = [h_s^1; \cdots; h_s^{L_e}]$. We use $o_s^{L_e}$ to compute the attention score in the proposed network. Acting in this essentially connects all the encoder layers with the decoder layer in attention computation. It still remains a question that how to use $o_s^{L_e}$ in the attention computing. One straightforward way is to replace $h_s^{L_e}$ with $o_s^{L_e}$ wherever it is used in Eqn. 4. Rather than that, we propose a more efficient way: to compute the attentive context vector $c_t$ in the $t$th decoding step, we use every part of $o_s^{L_e} = [h_s^1; \cdots; h_s^{L_e}]$, i.e., $h_s^l, \forall l \in \{1, \cdots, L_e\}$, to separately compute the context vectors $c_t^l$, and concatenate all of them to form $c_t$. Mathematically speaking, for each encoder layer $l$, we have:

$$e_{st}^l = align(s_{t-1}^{l'}, h_s^l), \alpha_{st}^l = \frac{\exp(e_{st}^l)}{\sum_{n=1}^N \exp(e_{nt}^l)}, c_t^l = \sum_{s=1}^N \alpha_{st}^l h_s^l, \tag{7}$$

where attention is computed at the $l'$-th layer of decoder. Then we concatenate all $c_t^l$ to form the final context vector $c_t$:

$$c_t = [c_t^1, c_t^2, \ldots; c_t^{L_e}]. \tag{8}$$

Acting in this way brings two advantages: 1) It achieves multi-branch and aggregated transformation in attention computation (i.e., rather than directly computing $c_t$, it first computes each subpart of $c_t$ and then concatenates them), which significantly reduces computation effort. Such multi-branch mechanism can obtain accuracy improvement as observed in modern CNN architectures such as Inception and ResNext (Szegedy et al., 2015; Xie et al., 2016); 2) All the encoder layers contribute to the attention computation by forming their own attentional weights ($\alpha$) and context vectors ($c$), forming an attention hierarchy which will be more expressive.

## 4 EXPERIMENTS

In order to demonstrate the effectiveness of our proposed DenseRNN, we conduct the empirical study on a variety of S2S benchmark tasks, including neural machine translation and text summarization. LSTM is used as basic recurrent unit and the attention models are the same with Bahdanau et al. (2014). We are primarily interested in whether DenseRNN can achieve better accuracy under similar number of parameters, or reach comparable accuracy with fewer number of parameters and faster convergence speed.

### 4.1 EXPERIMENT SETUPS

#### 4.1.1 DATASETS

We conduct experiments on two major machine translation tasks and a text summarization task.

**WMT-14 English-French** We use a subset of WMT-14 English-French dataset, which contains $12M$ parallel sentence pairs as training set. We use the concatenation of *newstest2012* and *newstest2013* as validation set. We use *newstest2014* as the test set. We pre-process words into subword units using byte pair encoding (BPE) (Sennrich et al., 2016) and use $30,000$ most frequent BPE tokens as the vocabulary for both English and French.

**WMT-14 English-German** We use the same setup as Luong et al. (2015) which contains $4.5M$ English-German sentence pairs. We use the concatenation of *newstest2012* and *newstest2013* as validation set and use *newstest2014* as test set. The vocabulary consists of $32,000$ sub-word units also processed with BPE.

**Text Summarization** For the text summarization experiment, we use the same data set as Wu et al. (2017), which is from a subset of Gigaword Corpus (Graff & Cieri, 2003). The source input is the first sentence of a news article and the target output is the headline of the article, as described in Rush et al. (2015) and Wu et al. (2017). There are $42212$ words in the source dictionary and $19014$ words in the target dictionary after processing the data in the same way as above works. Other words are replaced by a special token ('UNK'). We limit the maximum length of the sentence to $150$.

Table 1: Model configurations and BLEU scores on WMT-14 English-French translation task. We compare the baseline model with two variants of our proposed model, each with different layers and parameters.

| Model | Hidden size | # Layers | # Param | Valid BLEU | Test BLEU |
|---|---|---|---|---|---|
| baseline | 1024 | 4 | 230M | 32.20 | 38.40 |
| DenseRNN-10L | 256 | 10 | 211M | **32.57** | **38.91** |
| DenseRNN-6L | 256 | 6 | 117M | 32.31 | 38.50 |

Table 2: Model configurations and BLEU scores on WMT-14 English-German translation task. We compare the baseline model with two variants of our proposed model, each with different layers and parameters.

| Model | Hidden size | # Layers | # Param | Valid BLEU | Test BLEU |
|---|---|---|---|---|---|
| baseline | 1024 | 4 | 234M | 23.87 | 24.15 |
| DenseRNN-10L | 256 | 10 | 214M | 23.95 | **24.35** |
| DenseRNN-6L | 256 | 6 | 120M | **24.09** | 24.00 |

### 4.1.2 MODELS

**Machine Translation** For machine translation tasks, the baseline model is a 4-4 layer encoder-decoder model build with stacked LSTM (Bahdanau et al., 2014), with LSTM hidden dimension set as 1024. We build DenseRNNs with different number of layers, i.e., with different number of parameters, to compare their performance. As shown in Table 1 and Table 2, 'Beseline' model refers to the baseline stacked LSTM model. "DenseRNN" with suffix "10L" and "6L" refer to the two variants of our proposed dense RNN model, with same hidden size 256 but different layer numbers (i.e., 10 and 6). Both the word embedding sizes for stacked LSTM and our proposed DenseRNN are 512.

**Text Summarization** For text summarization, the baseline model is the same stacked LSTM model as in machine translation task. We build two baseline models with layer 4 and 6. For each baseline model, we build dense models with different number of layers, with comparable or half number of parameters, as shown in Table 3. Particularly, to investigate whether dense connection or dense attention contributes to the improvement, we build models with only dense connection (referred as DenseRNN(DC) where DC means dense connection) and models with both dense connection and dense attention (referred as DenseRNN(DC+DA) where DA means dense attention) and keep them in similar number of parameters.

### 4.1.3 TRAINING

**English-French translation** We train all the baseline models and DenseRNN models using Adadelta (Zeiler, 2012) with initial learning rate 1.0 and a learning rate annealing strategy that automatically divides the learning rate by 10 if the validation performance does not increase over an epoch. We also apply dropout (Srivastava et al., 2014) to the models. We use separate dropout rates for LSTM hidden states and the states before softmax layer, with both values set via validation set performance. We train all models on two Titan Xp GPU cards with mini-batch size 128.

**English-German translation and Text summarization** The training settings in both tasks are almost the same with English-French translation, except that training is conducted on a single Titan Xp GPU card with batch size 48 and 64 for English-German and text summarization task respectively.

## 4.2 RESULTS

For translation tasks, we use beam search (Sutskever et al., 2014) to decode target side sequence with beam width 6 following Luong et al. (2015) and evaluate the translating accuracy with tokenized

Table 3: Model configurations and ROUGE scores on Gigaword text summarization task. We build two baseline models and several our proposed dense models, each with different layers and parameters. DC means dense connection and DA means dense attention. R-1, R-2 and R-3 stand for ROUGE-1, ROUGE-2 and ROUGE-L score respectively.

| Model | Hidden | #Layers | #Param | R-1 | R-2 | R-L |
|---|---|---|---|---|---|---|
| baseline-4L | 512 | 4 | 71M | 35.91 | 16.68 | 33.01 |
| DenseRNN(DC)-4L | 128 | 4 | 41M | 36.10 | 17.04 | 33.53 |
| DenseRNN(DC+DA)-4L | 150 | 4 | 43M | 36.15 | 17.12 | 33.57 |
| DenseRNN(DC)-8L | 128 | 8 | 76M | 36.41 | 17.36 | 33.70 |
| DenseRNN(DC+DA)-8L | 142 | 8 | 78M | **36.48** | **17.37** | **33.72** |
| baseline-6L | 512 | 6 | 96M | 36.32 | 17.16 | 33.60 |
| DenseRNN(DC)-6L | 128 | 6 | 56M | 36.23 | 17.24 | 33.61 |
| DenseRNN(DC+DA)-6L | 144 | 6 | 58M | 36.32 | 17.16 | 33.67 |
| DenseRNN(DC)-10L | 128 | 10 | 100M | 36.50 | 17.41 | 33.65 |
| DenseRNN(DC+DA)-10L | 138 | 10 | 101M | **36.52** | **17.52** | **33.93** |

case-sensitive BLEU (Papineni et al., 2002)[1]. The results for WMT-14 English-French and English-German are shown in Table 1 and Table 2 respectively.

When comparing models with similar number of parameters, we can observe that DenseRNN-10L outperforms stacked RNN baseline on both English-French and English-German tasks, by a $0.51$ and $0.20$ margin in terms of BLEU. Furthermore, with only half number of parameters, DenseRNN-6L obtains similar performance with stacked RNN. This demonstrates that the dense connections in both encoder/decoder and attention help the information flows and encourage feature reuse as well as new feature exploration, resulting in more efficient parameter usage.

For text summarization task, we use decoding beam width 10 following Wu et al. (2017) and evaluate the model performance in terms of ROUGE-1, ROUGE-2 and ROUGE-L in Table 3. Similar experimental results are observed: comparing DenseRNN(DC+DA)-4L with baseline-4L, we can see that DenseRNN even outperforms baseline on ROUGE-1, ROUGE-2 and ROUGE-L score with gains of $0.24$, $0.54$ and $0.56$ respectively, but using only $43M$ parameters, around half of the number of parameters that baseline used ($71M$), which further verifies the efficiency of our proposed model. Moreover, from the results, we find that both dense connection and dense attention contribute to the information flow efficiency, as all the DenseRNN(DC) outperform corresponding baseline and DenseRNN(DC+DA) outperform corresponding baseline and DenseRNN(DC) when having similar number of parameters.

### 4.3 TRAINING TIME ANALYSIS

From the experiment results in last subsection, we observe that dense connections make more efficient usage of parameters. To figure out how it will influence the convergence speed and training time, we plot the training curves showing the validation set performance varying with the wall-clock time on the two translation tasks in Fig. 3 and Fig. 4. From the two figures, it is obvious that DenseRNN converges faster than stacked RNN baseline. For example, in Fig. 3, it takes stacked RNN roughly $1300,000$ seconds to achieve a BLEU score of 32.0 on validation set, while DenseRNN-10L uses roughly $1070,000$ seconds. Furthermore, DenseRNN-6L uses about only $630,000$ seconds, less than half the time of stacked RNN baseline.

## 5 RELATED WORK

There are several important factors that lead to the success of sequence-to-sequence (S2S) learning (Cho et al., 2014b; Sutskever et al., 2014; Bahdanau et al., 2014) in many real-world applications. For example, the usage of deep neural networks, whether RNN (Hochreiter & Schmidhuber,

---

[1]Calculated with *multi-bleu.perl* at `https://github.com/moses-smt/mosesdecoder/blob/master/scripts/generic/multi-bleu.perl`.

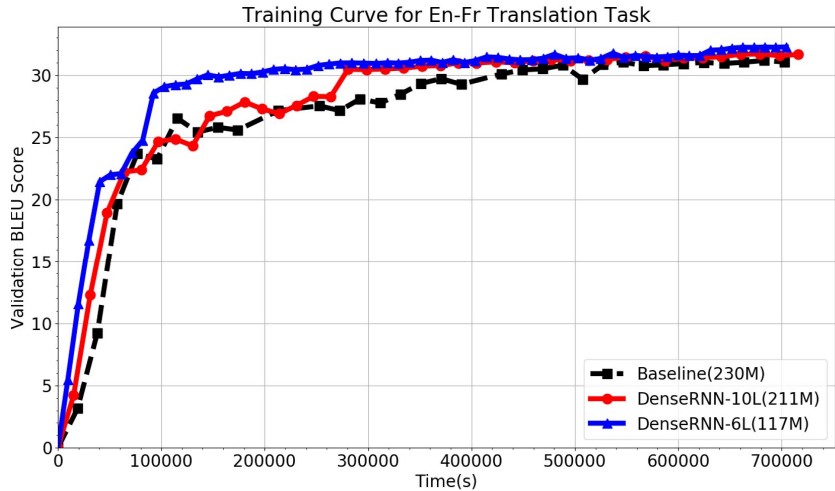

Figure 3: Performance on validation set with respect to training time in English-French translation task. All the models are trained on two Nvidia Titan Xp GPU cards.

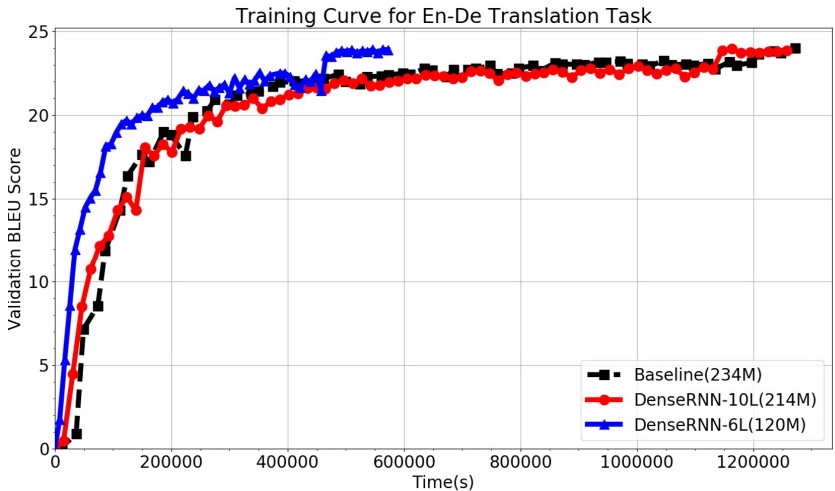

Figure 4: Performance on validation set with respect to training time in English-German translation task. All the models are trained on a single Nvidia Titan Xp GPU card.

1997) or CNN(Gehring et al., 2016; 2017) has made it to effectively leverage huge amount of parallel corpus in an end-to-end way. Furthermore, the attention model (Bahdanau et al., 2014; Luong et al., 2015; Mnih et al., 2014; Xu et al., 2015) in S2S learning effectively captures the long range dependency, by capturing the dynamic dependency within source and target side sequences. There are also very recent works relying on pure attention networks to achieve state-of-art performances in S2S learning (Vaswani et al., 2017).

Among all these S2S models, recurrent neural network (RNN) has been the initial architectures (Cho et al., 2014b; Sutskever et al., 2014; Bahdanau et al., 2014) and thus the most widely-adopted one in real-world applications. Powered by stacked architectures, one of the most typical deep RNN structures (Pascanu et al., 2013; Zhang et al., 2016; Zilly et al., 2017), S2S has achieved very promising results (Wu et al., 2016; Zhou et al., 2016; Wang et al., 2017) in neural machine translation.

We leverage the dense connections in DenseNet (Huang et al., 2016), one of the most successful deep convolutional neural network (CNN) architectures (Krizhevsky et al., 2012; Szegedy et al., 2015; He et al., 2016) for image recognition, to improve the performance of stacked RNN in S2S learning. By connecting every two layers in a deep CNN via concatenation operator, DenseNet achieves much better parameter usage efficiency, e.g., DenseNet with half of the parameters achieves similar ImageNet classification accuracy with ResNet (He et al., 2016).

## 6 CONCLUSION

In this work, we have designed densely connected recurrent neural networks for sequence-to-sequence learning, in which dense connections are applied to both the encoder/decoder and the attention mechanism to enhance information flow and feature reuse. Experiments on machine translation and text summarization have shown that our proposed model is efficient in terms of both parameter usage (memory usage) and training time.

As future work, we plan to explore deeper RNN models with the help of dense connections, and apply dense connections to other sequence-to-sequence learning models, like Transformer (Vaswani et al., 2017).

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
