# OpenReview forum: "DENSELY CONNECTED RECURRENT NEURAL NETWORK FOR SEQUENCE-TO-SEQUENCE LEARNING"
_ICLR.cc/2018/Conference — Reject_

### Official Review · AnonReviewer1 · 2017-11-26
**Dense skip-connections between layers improve recurrent networks**

**Rating:** 5
**Confidence:** 4

**Review:**

The article proposes to use dense skip-connections on the "vertical" (between-layers) connections of recurrent networks. Moreover, the article proposes to use separate attention-heads that run on the outputs of each encoder's layer, with each attention selecting other regions in the input to attend to.

The experiments demonstrate that the changes yield small BLEU score improvements on translation and summarization tasks.

I am not convinced by the presented results for the following reasons:
1) the paper introduces two concepts - the dense skip-connections and the multi-head attention. Experiments only show their joint impact, yet claims are made about the effectiveness of the skip-connections - maybe what's helping is the multi-head attention?
2) the results suggest that deeper model are better, with the densely connected networks being up to twice deeper than the baselines. What happens for deeper and narrower baselines that have a similar number of parameters?
3) looking at the training curves (thanks for including them), the densely connected model seems to converge faster by annealing the learning faster (I treat the "jumps" in the training curves as signs of learning rate anneal). Maybe this is what helps? I know the authors use an automaton to anneal the learning rate, but maybe the impact of learning rates should be evaluated?

Quality:
Good

Clarity:
The paper is clearly written.

Originality:
The addition of dense connections to recurrent networks is trivial.


Pros&cons
+ the proposed additions (dense skip connections) and multi-head attentions yield performance improvements
- the impact of the two contributions is not disentangled in the paper
- the two contributions are fairly obvious

---

> ### Author Response · Authors · 2018-01-02
> **New baseline method exploration and clarifications for training curve 'jump'**
>
> Dear reviewer,
> Thank you for your comments and questions. We would like to make several points for the sake of clarifications:
>
> (1) [Regarding the contribution about dense connections and dense attention]
>   Please refer to our general response, point 2.
>
> (2) [Regarding deeper and narrower baseline models]
>   To verify the effects of deep and narrow models V.S. shallow and fat models with roughly the same model capacity, we build another deeper and narrower baseline model with 22 layers, hidden size of 256, and 92M parameters in total (referred as baseline-22L), to compare with the baseline-4L (with 4 layers, hidden size of 512 and 96M parameters in total) we originally reported in the paper. The performance of baseline-22L is 35.71, 16.69 and 33.16 on ROUGE-1, ROUGE-2 and ROUGE-L respectively, which is similar to the shallow and fat baseline-4L with scores of 35.91, 16.68 and 33.01. It shows that baseline models obtain similar performance when having similar number of parameters, whether they are shallow and fat, or deep and narrow.
>
> (3) [Regarding the training time curve]
>   The "jump" is indeed caused by the learning rate annealing. Here what we want to specially make clear is that we make fair comparisons: the learning rate anneal strategy is the same for all the experiments including both baseline methods and our proposed methods: the model is evaluated on valid data set every 10000 steps and if the valid performance does not increase for 8 consecutive check points (in another word, 80000 steps) the learning rate will decay to 1/10.  Thus the "early jumps" indicate that the model has converged here under current learning rate and has to be fine-tuned. Therefore, "early jumps" not only indicate learning rate anneal, but more importantly, also indicates the speed of model convergence.

---

### Official Review · AnonReviewer3 · 2017-11-27
**incremental to prior work**

**Rating:** 6
**Confidence:** 4

**Review:**

This work proposes to densely connected layers to RNNs by concatenating previously constructed layers together as an input to the current layer. In addition, attention context is computed for each layer, then, combined together as a single context. Experimental results on English-French and English-German translation tasks and text summarization show comparable performance to a conventional non-densely connected layers with few number of parameters.

Motivation is clear in that it applies the densely connected networks in vision to texts and the gains achieved by smaller number of parameters look reasonable. However I have some concerns to this paper.

- It is a combination of two techniques, dense connections and multiple attention and it is not clear where the actual gain come from. I'd expect more ablation studies by isolating the effects of dense connection and the use of multiple attention mechanisms.

- It is not clear why the experiments for dense sticked to a particular hidden size, e.g., 256 for machine translation, and varies only the number of layers. Do you have experiments by fixing the number of layers and varying the hidden size?

Other comment:

- Section 3: sequence-to=sequence -> sequence-to-sequence

- It is not clear why the concatenation of all layers is not experimented which is mentioned in section 3.2. Memory problem?

---

> ### Author Response · Authors · 2018-01-02
> **Thank you for your reading and comments.**
>
> Dear reviewer,
> Thank you for your reading and comments. Below are the responses to your questions of the paper:
>
> (1) [Regarding your first concern on whether dense connection or dense attention actually makes the gain]
>   Please refer to point (1) of our general response to all reviewers. To state again, we conduct ablation studies on text summarization task and the results indicate that both dense connection and dense attention contribute to the improvement.
>
> (2) [Regarding model layer and hidden size]
>   We now add experiments of fixing layer number but varying hidden size on DenseRNN. We mainly focused on the parameters used, so we fixed the hidden size and varied layer number for convenience. As you concerned about this, we build dense models with fixed layer number but different hidden sizes to investigate its effects to performance. Concretely, we build four 4-layer dense models with hidden size of 150, 192, 256 and 320, leading to respectively 43M, 50M, 70M and 92M total model parameters, for the sake of comprehensive study.
>   We report the results here:
>   +---------------------------- +------------------+------------------+------------------+------------------+------------------+------------------+
>   |       Model                   | Hidden size  |     #Layers    |    #Param     |    ROUGE-1   |   ROUGE-2    |   ROUGE-L     |
>   |  DenseRNN-4L-h150|        150          |           4          |        43M        |        36.15      |        17.12      |        33.57       |
>   |  DenseRNN-4L-h192|        192          |           4          |        50M        |        36.25      |        17.28      |        33.56       |
>   |  DenseRNN-4L-h256|        256          |           4          |        70M        |        36.49      |        17.40      |        33.74       |
>   |  DenseRNN-4L-h320|        320          |           4          |        92M        |        36.56      |        17.57      |        33.78       |
>   +---------------------------- +------------------+------------------+------------------+------------------+------------------+------------------+
>  Compare these results with the results in Table 3, we can see that when fixing layer and varying hidden size to vary the model size, same observations as in the paper can be obtained: dense models show similar or even better performance as baselines when using only half parameters, and outperform baselines under similar number of parameters.
>
> (3) [Regarding your question of why the concatenation of all layers in section 3.2 is not experimented]
>   We respectfully disagree with your claim. Actually, we indeed did the same in experiment as in section 3.2. Section 3.2 describes the dense connection, where "dense" means that the attention operation is applied between hidden states in decoder and every layer`s output in encoder, and then all the context vectors are concatenated.

---

### Official Review · AnonReviewer2 · 2017-11-27
**review of "Densely connected recurrent neural network for sequence-to-sequence learning"**

**Rating:** 4
**Confidence:** 4

**Review:**

This paper describes an attempt of improving information flow in deep networks (but is used and tested here with seq2seq models although it is reality unrelated to seq2seq models per se). Slightly different from Resnet the information flow is improved by not just adding the outputs from previous layers but instead concatenating the outputs from previous layers with the current outputs. The authors claim better convergence speed and better results for a similar number of parameters although the differences seems to be in the noise.

Overall this is an OK technique but in my opinion not really novel enough to justify a whole paper about it as it seems more like a relatively minor architecture tweak. The results seem to indicate that there were some problems with getting deeper networks to work for the baseline (why is in Table 3 baseline-6L worse than baseline-4L?) for which the reason could be a multitude of issues probably related to hyper-parameter tuning. What is also missing is a an analysis of the negative consequences of this technique -- for example, doesn't the number of parameters increase with the depth of the network because of the concatenation? Also, it would have been good to see more experiments with smaller baseline networks as well to match the smaller DenseNet networks in Table 1 and 2. Finally, the writing of the paper could be improved a lot: The basic idea is not well described (however, many times repeated) and the grammar is often wrong and also there are some typos.

---

> ### Author Response · Authors · 2018-01-02
> **We really appreciate your constructive comments**
>
> Dear reviewer,
>
> We really appreciate your constructive comments and questions of the paper. Here are our responses to your questions:
>
> (1) [Regarding the significance of dense operation in deep neural networks for sequence-to-sequence learning]
>   It is well motivated to explore the potential of leveraging dense connections in sequence-to-sequence learning: 1) Sequence to sequence learning used to leverage single layer model, but recent work finds it necessary to use deep models for the sake of satisfactory, even stat-of-art performances. Tasking three most representative and effective sequence-to-sequence models as example: GNMT (Google Neural Machine Translation) (Wu et al., 2016) has 8 layers; ConvS2S(Convolutional Sequence to Sequence Learning) (Gehring et al., 2016) by Facebook uses 15 layers; The latest STOA sequence-to-sequence learning model – Transformer (Vaswani et al., 2017) - uses 6 layers as its basic model structure. However, each layer contains 4 sub-layer, leading to total 24 layers in both the encoder and decoder. Therefore, it is important to improve deep sequence-to-sequence models for the real-world usage.  2) Dense connections have been shown to be an effective approach in computer vision to improve parameter usage in building deep models. In terms of better performance/parameter usage, it is then a natural question that whether such dense operators help to improve the deep structures in sequence-to-sequence models such as stacked LSTM. 3) However it is not trivial to bring the dense operations to sequence-to-sequence learning: on one hand the model architectures (e.g., LSTM) are quite different with CNN; on the other hand, we need to specially handle the inter connections between encoder and decoder within the model. Out of all these items, we motivate the exploration of leveraging dense connections in sequence-to-sequence learning.
>
> (2) [Regarding the performance issue of deeper model that in Table 3 baseline-6L is worse than baseline-4L]
>   Please refer to point (1) of our general response to all reviewers.
>
> (3) [Regarding the number of parameters increase with the depth of the network]
>     When we do concatenation, since the information flow is enhanced, we do not need to use the hidden size as big as in stacked RNN so we decrease the hidden size much and thus decrease the total parameters.
>   We have shown the number of parameters of all the dense models in all our tables (Table 1, Table 2 and Table 3) in the experiments section. It shows our dense models use less parameter size.
>
> (4) [Smaller baseline models to match the smaller dense networks]
>   Thanks for your suggestion and we have conducted such new experiment. To be more concrete, on WMT English->German translation, to match the model size of our previously reported DenseRNN-6L with 117M parameters (test set BLEU score of 24.00 shown in Table 2), we additionally run a 2-layer baseline stack LSTM model with 133M parameters.  The test set BLEU of such a smaller baseline model is 23.56, worse than our DenseRNN-6L-117M (24.00).
>
> (5) [Regarding paper writing]
>   Thanks for your comments on the paper writing. We have fixed some typos and will further carefully polish the paper and improve the writing.

---

### Author Response · Authors · 2018-01-02
**Towards results on text summarization and ablation study**

Dear Reviewers,

We thank all of your constructive review comments which help us improve the paper. Here are some general points for some of your common issues:

(1) In the original version of the paper, due to limited time and resources, we apologize for not fully exploring the potential of different hyper parameter values of text summarization task, leading to a not satisfactory baseline method performance (e.g., the 6-6 layer baseline model performs worse than the 4-4 model).  Therefore we re-run all the experiments for this task with a comprehensive fine-tuning strategist for hyper-parameter values and achieve better performance for baseline models and our proposed models. Please refer to Table 3 for latest detailed results on text summarization.

(2) We regret for not investigating different contributions of dense connection in RNN layers and dense attention. In the latest paper version, we conduct such ablation studies using text summarization as an example and list the results in Table 3.  Overall speaking, we show that both dense connections and dense attention make contribution to the performance improvement and please check the analysis in the main text under Table 3 (section 4.2, paragraph 3).

---

### Decision · Program_Chairs · 2018-01-29
**ICLR 2018 Conference Acceptance Decision**

**Decision:**

Reject

**Comment:**

There is a general consensus that the novelty is too limited, as it is a straightforward combination of two simple and well-known ideas (dense skip connections / multi-head attention).

Pros:
-- the modifications yield (moderate) improvements in performance
Cons:
-- the modifications are relatively trivial, basically a combination of 2 well-known ideas

The authors addressed some of the reviewers concerns (e.g., disentangling impact of the two modifications) but unfortunately I still do not see as appropriate for the conference.